# Remote Sensing of Seawater Temperature and Salinity Profiles by the Brillouin Lidar Based on a Fizeau Interferometer and Multichannel Photomultiplier Tube

**DOI:** 10.3390/s23010446

**Published:** 2022-12-31

**Authors:** Yuanqing Wang, Yangrui Xu, Ping Chen, Kun Liang

**Affiliations:** School of Electronics and Information Engineering, Huazhong University of Science and Technology, Wuhan 430074, China

**Keywords:** Fizeau interferometer and multichannel PMT, Brillouin scattering spectrum, temperature and salinity profiles, Brillouin lidar, remote sensing

## Abstract

Brillouin spectroscopy is a powerful tool to measure the water temperature and salinity profiles of seawater. Considering the insufficiency of the current spectral measurement methods in real-time, spectral integrity, continuity, and stability, we developed a new lidar system for spectrum measurement on an airborne platform that is based on a Fizeau interferometer and multichannel photomultiplier tube. In this approach, the lidar system uses time-of-flight information to measure the depth and relies on Brillouin spectroscopy as the temperature and salinity indicator. In this study, the system parameters were first optimized and analyzed. Based on the analysis results, the performance of the system in terms of detection depth and accuracy was evaluated. The results showed that this method has strong anti-interference ability, and under a temperature measurement accuracy of 0.5 °C and a salinity measurement accuracy of 1‰, the effective detection depth exceeds 40.51 m. Therefore, the proposed method performs well and will be a good choice for achieving Brillouin lidar application in seawater remote sensing.

## 1. Introduction

Oceans are closely related to human life and production as they produce oxygen and regulate the Earth’s climate; they are also vital to economic development in terms of shipping and other maritime activities [1,2,3]. Various environmental factors, such as the temperature and salinity, are important parameters that can help characterize the state of seawater. An accurate determination of the ocean temperature and salinity is the basis for the development and sustenance of maritime activities [4,5,6]. 

Lidar, as an active remote sensing detector, is now widely used in ocean temperature and salinity monitoring [7,8,9,10,11,12]. In recent remote sensing research, Brillouin lidar is an attractive topic [13,14,15,16,17]. In Brillouin lidar, ocean environmental factors are mainly retrieved by detecting spectral parameters related to Brillouin scattering (such as the frequency shift and linewidth) [18,19]. Ergo, the accurate Brillouin spectrum measurement method is the core technology for ocean Brillouin lidar. 

The spectral measurement method using a scanning Fabry–Perot (FP) interferometer was first used to gradually scan the energies at each frequency point in the spectrum to obtain a complete Brillouin spectrum [20,21,22]. The Brillouin frequency shift was extracted from the spectrum and used for seawater temperature measurements. The temperature measurement accuracy reached 3 K, proving the feasibility and application prospects for the Brillouin spectrum measurement of seawater information. However, this method requires a long time (approximately 10 min) to acquire the Brillouin spectrum, which is not conducive to the practical applications of lidar.

Dai et al. [23] proposed a spectral measurement method based on edge technology to obtain the Brillouin spectrum in real time. In edge technology [24], a narrowband filter with a steep edge is placed near the Brillouin spectrum. A small shift in the spectrum causes a significant change in the energy after passing through the edge filter, and the spectral frequency shift can be measured through the change in the energy. In this edge detection method, an I_2_ absorption cell is used as the filter. Because the absorption spectral line of I_2_ is determined by the inherent characteristics of iodine molecules, the corresponding wavelength/frequency of the absorption spectral line is fixed. In other words, the parameters of the lidar system, such as the laser frequency, should strongly match the frequency of the absorption spectral line, making the system less flexible. To solve this problem, Walther et al. [25,26] designed an excited-state Faraday anomalous dispersion optical filter (ES-FADOF) that can adjust the central frequency and line shape of the edge filter by controlling the magnetic field strength, making the selection of the system parameters more flexible. However, the scattering spectrum acquired using the edge method is incomplete, and only one parameter, namely the Brillouin shift, can be obtained. Its application to the synchronous measurement of multiple marine environmental elements is limited.

To quickly obtain a complete Brillouin spectrum to determine more spectral characteristic parameters, such as the Brillouin linewidth, and satisfy the requirements of multiparameter synchronous remote sensing, Shi et al. [27] proposed a spectral measurement method using a Fabry–Perot (FP) etalon and intensified charge-coupled device (ICCD). In this method, light passing through the FP etalon is separated at different frequencies in the spatial position with coherent interference, forming a 2D interference ring with equal inclination. The 2D image is then photoelectrically converted and outputted using an array ICCD. After subsequent image and signal processing, a 1D complete scattering spectrum can be obtained. Kun et al. [28,29] performed simultaneous measurements of seawater temperature and salinity based on multispectral characteristic parameters, such as the Brillouin frequency shift and linewidth, extracted from a complete scattering spectrum. However, because of the limitations of the integration time and frame rate (millisecond level) of the ICCD, the sampling rate of the Brillouin scattering spectrum is not high, and it is impossible to measure the vertical profiles of the temperature and salinity of high-density seawater at a vertical resolution of meters (time resolution of approximately 10 ns). In other words, this spectral measurement method fails to meet the profile measurement requirements for marine elements.

Considering the requirements of real-time spectrum measurement, spectral integrity, and continuity, Kun et al. [30,31] proposed a spectral measurement method based on a dual-edge etalon combined with a photomultiplier tube (PMT). In this method, the energies of two local characteristic narrowband spectra are measured through two edge filters. Because the function expression of the Brillouin scattering spectrum is known, a complete Brillouin scattering spectrum can be reconstructed using these two energies. With this method, a continuous Brillouin scattering spectrum is obtained, and rapid measurement of the temperature and salinity profiles can be realized with a vertical resolution of 1 m underwater (temperature and salinity accuracies of 0.5 °C and 1‰, respectively), further promoting the application of Brillouin lidar in the profile measurement of multiple marine environmental factors.

Compared with other Brillouin scattering spectral measurement methods, the measurement method based on the double-edge technology can satisfy the requirements of rapidity, spectral integrity, and profile continuity. However, a high environmental stability is required to maintain the stability of the central frequency and bandwidth of the dual etalon. A small change in the environment can change the instrument function of the etalon, resulting in a significant change in the energy passing through the filter, which affects the spectral reconstruction accuracy. An energy fluctuation error of 1% through the etalon can cause a scattering spectrum reconstruction error of 40 MHz, and the corresponding temperature error reaches 1 °C. Therefore, although this method can be used to reconstruct the Brillouin scattering spectrum with a high accuracy in theory, considering that the measured energy is prone to interference under environmental conditions, the detection accuracy of ocean environment parameters is limited in actual implementation. For example, this method is unsuitable for applications with higher accuracy requirements (a temperature measurement accuracy of 0.2 °C or higher). 

In order to solve the shortage of the measurement method based on the double-edge technology in high-precision detection, another method must be proposed. A theoretical analysis of the Brillouin scattering spectrum has shown that this spectrum is symmetrically distributed on the left and right sides at the center of the Rayleigh spectrum. If the Rayleigh spectrum is taken as the reference ruler, even if the system is unstable and there is jitter in the entire scattering spectrum, the relative positions of the Brillouin and Rayleigh spectra will not change. Therefore, if the complete Rayleigh and Brillouin spectra can be measured, the stability of the spectral measurement and the anti-interference ability can be improved, and the measurement results will be more accurate. 

Hence, this paper proposes a scattering spectrum measurement method based on a Fizeau interferometer and multichannel PMT. When backscattered light passes through the Fizeau interferometer, equal-thickness interference fringes are generated, and each fringe is the expansion of one complete Rayleigh–Brillouin spectrum in space. After beam shaping, 1D line array images can be outputted and received by a multichannel PMT that quantifies the continuous Rayleigh–Brillouin (RB) scattering spectrum into discrete sparse energy points [32,33]. Finally, the complete RB scattering spectrum is reconstructed based on these points. This method can solve the drawbacks of the double-edge technology in terms of stability and accuracy, and improve the measurement accuracy and detection depth of seawater parameter profiles when using the Brillouin scattering spectrum. 

In the remainder of this article, we introduce the theory and system of the Fizeau interferometer combined with the multichannel PMT method in the second part. The system parameter optimization and analysis are described in Section 3. In Section 4, we discuss the performance of the Brillouin lidar system. Finally, the conclusions are presented in Section 5. 

## 2. System and Theory

### 2.1. Brillouin Lidar System 

Figure 1 shows the schematic of the Brillouin spectrum measurement system based on the Fizeau interferometer combined with a multichannel PMT. The system can be divided into two parts: a light emission part and a spectral reception part. The light emitted from the pulsed laser is split into two. One beam with a low amount of light is set as the reference beam received by PMT0 to monitor the jitter in the laser energy. After expansion, the second beam enters the water to produce scattered light. In the spectral reception part, the 180° backscattered light is first collected by a Cassegrain telescope and transmitted through an optical fiber. After collimation, the light passes through a narrowband filter to further reduce the background light and simultaneously suppress the Raman-scattered light. Subsequently, the filtered scattering is yielded in the Fizeau interferometer. The Fizeau interferometer in the system was designed to allow only one interference fringe to appear. The interference fringe passes through a cylindrical lens to compress the light in the direction in which the light in the fringe has the same frequency. Finally, the compressed light is detected by the multichannel PMT, which converts the continuous scattering spectrum into discrete electrical signals. The system also includes a control circuit to control the pulse laser emission, read PMT data, enable real-time interaction through a computer, and control the temperature regulation of the Fizeau interferometer. Table 1 presents the designed parameters of the airborne platform system and typical environmental parameters.

Figure 2 shows the detection process of the RB scattering spectrum for the spectral reception part. 

The backscattered light passes through the Fizeau interferometer (FI). Under the principle of multibeam interference, light of different frequencies is expanded in space after FI, and a continuous RB scattering spectrum is formed and exactly distributed on the multichannel PMT after beam shaping. The continuous spectrum is then quantified by the multichannel PMT, and a discrete RB scattering spectrum is obtained. Based on the spectral reconstruction method, the discrete RB scattering spectrum was used to reconstruct the continuous RB scattering spectrum using a software algorithm. 

### 2.2. Theory of Measurement

Theoretically, the RB scattering spectrum of seawater is related to its temperature and salinity and can be expressed by three Lorentzian functions:(1)$$\begin{array}{c} f_{RB}\left(v\right)=f_{B}\left(v\right)+f_{R}\left(v\right)\\ =\frac{I_{B}}{\pi \Gamma_{B}}\left\{\frac{1}{1+{\left[2\left(v-v_{B}\right)/\Gamma_{B}\right]}^{2}}+\frac{1}{1+{\left[2\left(v+v_{B}\right)/\Gamma_{B}\right]}^{2}}\right\}+\frac{I_{R}}{\pi \Gamma_{R}}\cdot \frac{1}{1+{\left[2\left(v-v_{R}\right)/\Gamma_{R}\right]}^{2}} \end{array}$$
where *f_B_*(*v*) and *f_R_*(*v*) represent the Brillouin and Rayleigh components, respectively. $\Gamma_{B}$ is the Brillouin linewidth, $v_{B}$ is the Brillouin shift, v is the frequency, $I_{B}$ is the intensity of the Brillouin light, $v_{R}$ is the central frequency of the Rayleigh spectrum (the same as the laser center frequency), $\Gamma_{R}$ is the Rayleigh scattering linewidth, and $I_{R}$ is the intensity of Rayleigh scattering.

Generally, the RB scattering light after passing through the Fizeau interferometer can be expressed as the convolution of the RB scattering spectrum $f_{RB}\left(v\right)$ and the instrument function of the Fizeau interferometer *FI*(*v*):(2)$$f_{CL}=FI\left(v\right)⮾{\;f}_{RB}\left(v\right)$$
where
(3)$$\begin{array}{c} \mathrm{FI}\left(v\right){=I}_{0}\cdot \frac{1}{1+{\left(\frac{2\cdot \mathrm{FSR}}{{\pi \;\Gamma }_{\mathrm{FI}}}\right)}^{2}\cdot \sin^{2}\left(\frac{\pi }{\mathrm{FSR}}\cdot v\right)}\; \end{array}$$

Here, $I_{0}$ is the maximum intensity of the transmission spectrum, *FSR* is the free spectral range, and *Γ_FI_* is the half-height and full-width of the Fizeau interferometer.

In addition to the backscattered *RB* signal of seawater, the signal received after beam shaping contains background and detector noise, which can be computed as follows: (4)$$I_{0}\left(v\right)=FI\left(v\right)⮾\;\left[f_{RB}\left(v\right)+N_{M}+N_{b}\right]+N_{d}$$
where *N_b_* is the background noise, *N_M_* is the groundscattering noise generated by the particles in water, and *N_d_* is the dark-current noise of the detector. Because the PMT outputs discrete voltage signals, the continuous signal *I*_0_(*v*) is discretized by the multichannel PMT. Assuming that the number of channels in the multichannel PMT is *N*, the energy received by each channel can be expressed as follows:(5)$$\begin{array}{cc} I_{i} & =f_{Ch}\left(v_{i},\Gamma_{i}\right)\;\cdot \;I_{0}\left(v\right)=\overset{v_{i}-\frac{\Gamma_{i}}{2}}{\underset{v_{i}+\frac{\Gamma_{i}}{2}}{\int }}I_{0}\left(v\right)dv\\ & =\int_{v_{i}+\frac{\Gamma_{i}}{2}}^{v_{i}-\frac{\Gamma_{i}}{2}}\left\{FI\left(v\right)⮾\;\left[f_{RB}\left(v\right)+N_{M}+N_{b}\right]+N_{d}\right\}dv \end{array}$$
where $I_{i}$ (*i* = 1, 2, …, *N*) represents the energy received by each channel, *f_Ch_* (*v_i_*, *Γ**_i_*) is the instrument function for each channel, *v*_i_ is the center frequency of each channel, and *Γ**_i_* is the bandwidth. A discrete array [*I*_1_, *I*_2_, …, *I_N_*] comprising $I_{i}$ represents the energy detected by the multichannel PMT.

Using this system, a discrete RB scattering spectrum can be obtained. The following step is for the additional Brillouin characteristic parameters based on the discrete array to retrieve the temperature and salinity. By fitting the discrete array [*I*_1_, *I*_2_, …, *I_N_*] using Equation (1):(6)$$S_{RB}\left(v_{B},\Gamma_{B}\right)=fit\left(\left[I_{1},\;I_{2},\;I_{3},\cdots ,\;I_{n}\right]\right)$$
the *RB* scattering spectra *S_RB_* (*v_B_*, *Γ_B_*) can be reconstructed, and the Brillouin shift and linewidth can be acquired. 

When substituting the Brillouin shift and linewidth into the dual-parameter inversion model, the corresponding temperature and salinity of the seawater can be calculated, and the measurement of the ocean environment parameter profile is realized. The synchronous retrieval models for the temperature and salinity can be expressed as follows:(7)$$\begin{array}{cc} T\left(v_{B},\Gamma_{B}\right) & =m_{1}+m_{2}\cdot v_{B}+\frac{m_{3}}{\Gamma_{B}}+m_{4}\cdot v_{B}{}^{2}+\frac{m_{5}}{\Gamma_{B}{}^{2}}+\frac{m_{6}\cdot v_{B}}{\Gamma_{B}}\\ & +m_{7}\cdot v_{B}{}^{3}+\frac{m_{8}}{\Gamma_{B}{}^{3}}+\frac{m_{9}\cdot v_{B}}{\Gamma_{B}{}^{2}}+\frac{m_{10}\cdot v_{B}{}^{2}}{\Gamma_{B}} \end{array}$$
(8)$$\begin{array}{cc} S\left(v_{B},\Gamma_{B}\right) & =k_{1}+\frac{k_{2}}{v_{B}}+\frac{k_{3}}{v_{B}^{2}}+\frac{k_{4}}{v_{B}^{3}}+\frac{k_{5}}{v_{B}^{4}}+\frac{k_{6}}{v_{B}^{5}}+k_{7}\ln\Gamma_{B}\\ & \;+k_{8}\ln\Gamma_{B}{}^{2}+k_{9}\ln\Gamma_{B}{}^{3}+k_{10}\ln\Gamma_{B}{}^{4}+k_{11}\ln\Gamma_{B}{}^{5} \end{array}$$
where the coefficient values of *m_i_* and *k_i_* can be referred from [34]. For this dual-parameter inversion model, the temperature and salinity inversion errors were within 0.08 °C and 0.21‰, respectively. 

As the accuracy of the dual-parameter inversion model is high, the main factor affecting the accuracy of the seawater temperature and salinity measurements is the accuracy of the Brillouin scattering spectrum measurement. The quality of the spectrum can be weighted using the SNR. For the lidar detection system using the PMT as the receiving detector, the SNR of the received signal is as follows:(9)$$\mathrm{SNR}=\frac{N_{B}\left(z\right)}{\sqrt{\beta \cdot N_{B}\left(z\right)}+N_{d}+N_{b}}\cdot \sqrt{M}$$
where *M* is the cumulative number, and *β* is the noise coefficient, which is generally between 1.2 and 2.0. Based on an analysis of existing experimental data [31], the value was chosen to be 2.0.

In summary, based on system optimization, this study mainly analyzed the measurement depth of the system, the accuracy of the Brillouin scattering spectrum measurement, and the accuracy of the retrieved temperature and salinity under different SNR conditions.

## 3. System Parameters Optimization and Analysis

The measurement accuracy of the Brillouin scattering spectrum for the system based on the Fizeau interferometer and multichannel PMT is mainly related to the spectral reception and light emission parts. For spectral reception, the parameters of the Fizeau interferometer and the channel number of the PMT significantly influence the accuracy of the results. For the light emission part, the laser frequency stability is the major factor. Therefore, system parameter optimization and analysis are discussed based on these three aspects. 

A Fizeau interferometer was used for the spectrum separation. Two parameters are important: the free spectral range (FSR) and the finesse. The FSR is the spacing in the optical frequency between two successive reflected or transmitted optical intensities. If the FSR is lower than that of the scattering spectrum, the spectrum after the Fizeau interferometer overlaps with its adjacent spectrum. Considering that the frequency range of the RB scattering spectrum of seawater is approximately 20 GHz [7,18], we set the FSR to 25 GHz. The finesse decides the width of the single-frequency light after the Fizeau interferometer. If the finesse is low, the spectrum after the Fizeau interferometer will be wide, corresponding to a low-frequency resolution. In contrast, a high finesse can produce a high spectral resolution. However, a high finesse reduces the amount of light transmitted after the interferometer. Less light implies a reduced detection distance. Hence, it is important to appropriately set the finesse. The finesse was determined based on the reflectivity of the Fizeau interferometer. From the analysis reported in [35,36], the reflectivity of the Fizeau interferometer was set to 90%.

For the PMT, the dark current and quantification efficiency cannot be adjusted after it is produced. The measurement accuracy of the Brillouin scattering spectrum is primarily related to the number of channels in the PMT. During spectral reception, the number of channels in the multichannel PMT determines the number of discrete spectral points. As the channel number increases, the number of discrete points increases, and the accuracy of the reconstructed spectrum increases. However, a higher number of channels implies that the energy obtained by each channel is lower, which reduces the SNR, and the spectrum is less precise. Therefore, it is necessary to appropriately select the number of channels for the PMT.

To analyze the relationship between the channel number of the PMT and the measurement accuracy, simulations of the Brillouin spectrum for different channel numbers were performed. The simulations were made based on the parameters listed in Table 1 using Equations (1)–(4). Scattering spectra of seawater at different PMT channel numbers were obtained, as shown in Figure 3. In the simulations, a typical condition with a water temperature of 20 °C and a salinity of 5‰ was selected. For effective signal detection, the SNR should be greater than 3. Considering the system margin, Poisson noise from the signal was added during the simulation with an SNR of 30. 

Using the discrete spectra, the corresponding continuous spectra were reconstructed based on Equation (4). The Brillouin shift and linewidth obtained from these continuous spectra were extracted. Figure 4 shows their measurement accuracies under different channel numbers of the PMT.

As it can be seen from Figure 4, when the channel number is over 12, both the deviations of the Brillouin shift and linewidth are less than 2 MHz. Moreover, the more channels, the higher the spectral inversion accuracy, and the higher the corresponding seawater temperature and salinity measurement accuracy. However, more PMT channels mean that the system is more complex and costlier. Except that, the data processing algorithm is more complex and the time complexity is greater. Considering the existing productions of PMT and cost, as well as the results for the simulations we did with SNR below 30, the channel number of 16 for the PMT is selected in this system.

For the light emission part, the instability of the light source causes error in the measurement results. As the energy changes can be eliminated by the reference beam, the laser frequency jitter is mainly discussed here. The laser frequency jitter makes the spectrum received by the multichannel PMT produce a shift that influences the accuracy of the retrieved Brillouin shift and linewidth. Therefore, the effect of the laser frequency jitter is discussed here. The interval between the centers of two adjacent PMT channels is 1 mm [37], which corresponds to the frequency of 0.762 GHz. It is only necessary to consider the spectral offset within 1 mm, as the offset exceeding 1 mm can be converted to less than 1 mm. The measurement errors of Brillouin shift and linewidth when the overall spectral drift is 1 mm are shown in Figure 5.

Figure 5 shows that when the laser or other factors cause spectral shift, the Brillouin shift error is less than 2 MHz; the Brillouin linewidth error is mostly less than 3 MHz. This demonstrates that the RB spectra measurement method has strong anti-interference capability.

## 4. Brillouin Lidar System Performance

In the last section, we optimized and analyzed the Brillouin lidar system. In this section, we discuss its performance. To analyze the measurement performance of the system, the depth detection ability was first analyzed. To analyze the actual depth detection capability of the system, a Brillouin energy measurement experiment was conducted. Using the measurement results, the depth deception of the system was explored through an equivalent analogy. In addition, the measurement accuracy is an important aspect of the measurement performance of the system. The temperature and salinity measurement accuracies and effective detection distance of this method under noisy conditions were evaluated. The detection performance of this method under different depth conditions was analyzed, and the maximum measurement depth was determined under a given temperature and salinity measurement accuracy.

### 4.1. Detection Depth Analysis of the Lidar System

The detection ability of a lidar system is related to the echo energy. For spontaneous Brillouin scattering, the backscattering energy can be quantified by the number of Brillouin scattering photons *N_B_*(*z*) in water, which can be expressed by the following equation:(10)$$N_{B}\left(z\right)=\frac{E\lambda }{hc}\cdot \frac{c\tau }{2n}\cdot \frac{\pi r^{2}}{{\left(z+nH\right)}^{2}}\cdot \alpha_{A}\cdot e^{-\left(2\alpha z\right)}\sigma \xi_{s}\xi_{l}\;$$
where *z* is the depth under water, *c* is the speed of light in vacuum, and σ is the backscattering coefficient, which comprises the Rayleigh scattering coefficient *σ*_R_ and Brillouin backscattering coefficient *σ*_B_.

#### 4.1.1. Experiment of Brillouin Energy Measurement

O’Connor et al. [38] proved that the energy ratio of Rayleigh scattering and Brillouin scattering in pure water at temperatures in the range of 0–40 °C is less than 0.04, which means that the Rayleigh light has a significantly lower intensity than the Brillouin light in water. However, in actual measurements, water contains various impurities, that is, the received scattering signals in water include Rayleigh scattering, Brillouin scattering, and Mie scattering signals. Because the Brillouin scattering peaks are on both sides of the Rayleigh and Mie scattering peaks in the spectrum, the intermediate Rayleigh and particle scattering signals can be filtered to obtain the pure Brillouin scattering signal. Figure 6 shows the optical design. The backscattered light was collected by a collecting optics. After filtering the background noise in the scattering spectrum, the light was split into two using a beam splitter. One beam was received by PMT2 and used as the reference beam, while the other passed through an absorption filter (iodine molecule absorption cell) to remove the unshifted central Rayleigh and Mie scattering light. Finally, pure Brillouin light was received by PMT1.

With this design, a setup for the Brillouin energy measurement was built. Figure 7 shows the schematic of the experimental system.

In this Brillouin energy measurement system, a parallel-axis transceiver design was adopted. The laser emitted pulses into the water, and the telescope composed of L1 and L2 received the scattered signals. The received scattered signal was divided into two beams using a 1:1 beam filter. One beam was directly received by PMT2, which was the total energy of the scattering signal. The other beam, which was the Brillouin scattering energy, was received by PMT1 after passing through the iodine molecular absorption cell. An iodine molecule absorption cell can absorb the central Rayleigh and Mie scattering light in water. Figure 8 shows the experimentally measured absorption curves of the iodine cells at different temperatures.

Figure 8 shows the iodine absorption lines at temperatures ranging from 30 to 65 °C, with a frequency of 0 Hz corresponding to the center frequency of the laser. The higher the temperature, the more evident the inhibition effect of the iodine molecular absorption cell on the Rayleigh–Mie scattering signal, but accordingly, the stronger the absorption ratio of the Brillouin scattering signal. To remove the Rayleigh–Mie scattering signal as much as possible and retain more of the Brillouin scattering signal, a temperature of 40 °C was set in the experiment, and the transmission efficiency of the Brillouin scattering signal was approximately 80%. Table 2 presents the system parameters used in the experiment.

When the water temperature was 28.2 °C and the salinity was 0‰, the scattering signals before and after passing through the iodine cell were collected, as shown in Figure 9.

Figure 9a,b show the measurement signals in front of the iodine cell and after the iodine cell, respectively. After averaging 100 times, we calculated the SNR to be 7.4, which was high.

Figure 10 shows the results of the Brillouin scattering energy and total scattering energy after averaging 100 times. The red and blue lines indicate the distributions of the signal energy received before and after passing through the iodine cell, that is, the total scattering energy and Brillouin scattering energy, respectively. The maximum values of the red and blue lines are 0.47 and 0.077, respectively, with a corresponding energy ratio of Brillouin scattering, and the total scattering energy is 16.38%.

The signal from 17 to 18 m at the tail of the signal in Figure 10 does not contain a scattered signal; this signal is mainly the dark-current noise of the detector. The mean value and variance of the dark-current noise were denoted by $\bar{X}$ and δ, respectively. When subtracting the mean value $\bar{X}$ from the received signal, the remaining signal that is greater than 3δ and has duration of more than 5 ns is considered an effective signal, and the maximum measurement depth of the Brillouin scattering signal was found to be 13.68 m. When the effective signal was determined with a rest signal greater than δ and with duration of more than 5 ns, the maximum measurement depth was calculated as 16.04 m.

The signal received by the PMT is the scattering energy over time. This can be determined using Equation (10). When the parameters are selected, as listed in Table 2, the attenuation coefficient of water can be obtained by fitting the signal in the range from maximum to the bottom. Figure 11 shows the fitting results of the Brillouin scattering energy and total scattering energy.

The attenuation coefficients obtained by fitting the total scattering energy signal and Brillouin scattering energy signal are 0.091 and 0.108, respectively. The two fitting results are very close. The attenuation coefficient of water in this experiment was set as α = 0.099. Using this value and the parameters listed in Table 2, the number of Brillouin scattered photons at different water depths was analyzed based on Equation (10), as shown in Figure 12. The figure shows the relationship between the number of received scattered photons and the measurement depth at integration times of 1, 100, and 1000. The black and pink dotted lines in the figure correspond to depths of 13.68 and 16.04 m, respectively. The number of system integrations is 100, and the numbers of scattered photons corresponding to depths of 13.68 and 16.04 m are 111,169 and 55,700, respectively.

#### 4.1.2. Analogy of Maximum Detection Depth

For a Brillouin lidar with different system parameters, the corresponding detectable Brillouin spectrum level remains the same as long as the number of Brillouin scattering photons remains unchanged. Therefore, the detection depth can be deduced from one setup to the next. In the experiment, the minimum number of photons of Brillouin scattering that the PMT can detect was obtained. This result can be used to deduce the maximum detection range of the lidar system based on the Fizeau interferometer and multichannel PMT under the parameter conditions listed in Table 1.

The lidar equation, Equation (10), is used to make an equivalent analogy: for two different lidar systems, under the condition that the number of Brillouin scattering photons is the same, the maximum measurement depth under other conditions can be deduced by:(11)$$\{\begin{array}{c} N_{B}\left(z_{0}\right)=M_{0}\cdot \frac{E_{0}\lambda_{0}}{hc}\cdot \frac{c\tau_{0}}{2n}\cdot \frac{\pi r_{0}{}^{2}}{{\left(z_{0}+nH_{0}\right)}^{2}}\cdot \alpha_{A}\cdot e^{-\left(2{\alpha z}_{0}\right)}\sigma \xi_{s}\xi_{l}\;\\ N_{B}\left(z_{x}\right)=M_{1}\cdot \frac{E_{1}\lambda_{1}}{hc}\cdot \frac{c\tau_{1}}{2n}\cdot \frac{\pi r_{1}{}^{2}}{{\left(z_{x}+nH_{1}\right)}^{2}}\cdot \alpha_{A}\cdot e^{-\left(2{\alpha z}_{x}\right)}\sigma \xi_{s}\xi_{l}\; \end{array}$$
where *N_B_*(*z*_0_) and *N_B_*(*z*_x_) are the numbers of Brillouin photons obtained under the experimental conditions and by the designed system, respectively.

The maximum measurement depth of the experiment was *z*_0_ = 13.68 m. By substituting the parameters listed in Table 1 into Equation (11), we can obtain the maximum Brillouin scattering signal detection depth under the equivalent condition of *M*_1_ = 1000 as *z*_x_ = 42.90 m. Under different water quality conditions and cumulative numbers of measurement, the maximum measurement depths will be different, as shown in Figure 13. The results show that, as the attenuation coefficient increases, the maximum measurement depth decreases, and when adding the cumulative number of measurements, the detection distance increases.

### 4.2. Detection Accuracy Analysis of Lidar System

The detection accuracy of the Brillouin lidar in ocean environments is related to the spectral signal corresponding to Brillouin scattering. Theoretically, the greater the depth, the weaker the scattering echo signal energy and the lower the detection accuracy. To analyze the effective detection depth of the system, the detection accuracies of the temperature and salinity based on the system at different depths obtained from the SNR of the echo signal are discussed in this section.

#### 4.2.1. SNR and Measurement Depth Analysis

The detection depth of the system is not only affected by the strength of the received signal but also by the detected noise. In the measurement of the profiles of water environment elements by a Brillouin scattering lidar, the main noise sources are radiation light from external light sources, random particle noise of photons, dark current, and thermal noise of detectors. For a lidar system based on a Fizeau interferometer and 16-channel PMT, the noise sources mainly include the detector noise *N_d_* and scattering noise of the external light source *N_b_*, expressed as follows:(12)$$\{\begin{array}{c} N_{d}=CPS\cdot \Delta t\\ N_{b}=\frac{\eta \lambda }{hc}\cdot E_{b}\cdot \pi \cdot {\left(\frac{\theta_{r}}{2}\right)}^{2}\cdot \Delta t\cdot A\cdot T_{s}\cdot \Delta \lambda \end{array}$$
where *η* is the receiving efficiency of the detector, *θ_r_* is the receiving field angle of the telescope, *A* is the aperture area, *E_b_* is the energy of the external light source (*E_b_* = 0 at night), Δ*λ* is the half-height linewidth of the filter, and *T_s_* is the optical receiving efficiency of the detector.

Equation (9) shows that, the greater the number of accumulated pulses, the higher the SNR of the system; that is, the longer the integration time, the better the SNR of the system. However, integration for long periods limits the temporal resolution; therefore, the relationship between the SNR and the temporal resolution should also be comprehensively considered.

To analyze the measurement accuracy of the ocean environmental element profiles obtained by the system, a temperature of 20 °C and salinity of 5‰ was set as the condition for the simulation in this section. The relationship between the SNR and the measurement accuracy of a single-pulse measurement can be obtained by substituting the scattering echo signal with different SNRs into the Fizeau interferometer and multichannel PMT system, as shown in Figure 14.

Figure 14 shows that as the SNR increases, which corresponds to a decrease in depth, the results of the temperature and salinity measurements become increasingly accurate. For a temperature accuracy of 0.5 °C and salinity accuracy of 1‰, the SNR should be greater than 35. At this time, the effective detection depth of the single-pulse measurement was 27.73 m. The SNR of the scattered signal must be improved to increase the effective detection depth of the system.

In addition to increasing the laser energy and improving the system reception efficiency, the SNR can be improved by adding the number of photons detected by the PMT through the integration and accumulation of multiple measured signals. Therefore, we focused on analyzing the relationship between the number of echo photons in the signal and the measurement depth.

#### 4.2.2. Echo Photon Number and Detection Depth Analysis

The SNR of the echo signal not only determines the temperature and salinity measurement accuracies of the system but also determines its effective detection range. Under a temperature accuracy of 0.5 °C and salinity accuracy of 1‰, the effective detection SNR was 35, corresponding to a detection depth of 27.73 m for a single measurement. When the number of received photons was kept unchanged, the SNR remained unchanged, and the measurement accuracies of the temperature and salinity would not change. Therefore, when the SNR of the signal was set to 35 by means of cumulative integration, the temperature and salinity measurement accuracies were ensured to be 0.5 °C and 1‰, respectively. At this time, the depth corresponding to the photon number measured at a single time was the effective detection depth. To explore the effect of cumulative times, repeated measurement results under an SNR of 35 were simulated, as shown in Figure 15. The different colors in the figure represent the energy results after a 16-channel PMT receives the spectrum. The red line indicates the original RB spectrum. The results show that a single measurement value is random owing to the influence of noise. Through averaging, we can obtain more accurate multichannel PMT measurement results.

Theoretically, the number of scattered echo photons excited by a single laser pulse remains constant. Considering that the laser repetition frequency of this system was 100 Hz, typical integration times of 1, 10, and 100 s, which correspond to accumulation numbers of 100, 1000, and 10,000, respectively, were selected for analysis, as shown in Figure 16.

The number of received echo photons and SNR can be ensured to remain unchanged by means of cumulative integration. The results in Figure 16 show that under a temperature measurement accuracy of 0.5 °C and salinity measurement accuracy of 1‰ (the corresponding SNR is 35), the effective detection depths under accumulation numbers of 100, 1000, and 10,000 are 75.48, 97.71, and 119.28 m, respectively.

Under the condition of ensuring the same measurement accuracy, despite the longer integration time, the corresponding effective detection depth was greater. However, the time resolution was reduced with a longer integration time. Considering the requirements of the effective measurement depth and time resolution, we chose an integration time of 10 s for the system, that is, 1000 pulses were measured accumulatively.

When the ocean environmental parameters change, for example, when the water quality changes, the measurement depth of the lidar system varies. Here, the effective detection depth of the system is discussed for typical water attenuation coefficients of 0.05, 0.08, 0.1, and 0.2. Figure 17 shows the single-pulse measurement results for different water attenuation coefficients under an integration time of 10 s.

As shown in Figure 17, under the condition that the measurement accuracies of the temperature and salinity are 0.5 °C and 1‰, respectively, the effective detection depth is 152.10 m for an attenuation coefficient *α* = 0.5. When the attenuation coefficient *α* = 0.2, the effective detection depth is 40.51 m. The results show that for an accumulation time of 10 s, the effective detection depth with the measurement method involving the Fizeau interferometer and 16-channel multichannel PMT in water of various qualities exceeds 40.51 m.

In Section 4.1.2, Figure 13 shows that, after applying the equivalent analogy, the maximum detection range is greater than 20 m when the attenuation coefficient is 0.2, and the accumulation number is 1000. Compared with the theoretical maximum effective detection range of 40.51 m, the detection capability of the actual system is lower. This may be attributed to the fact that the system efficiency of the actual experimental system is lower than that of the theoretical simulation system (for example, the passing rate of the iodine molecular absorption cell at 40 °C is 0.8). In an actual system, considering the attenuation of the optical lens and the efficiency of the detector, the theoretical detection efficiency cannot be achieved.

## 5. Discussion

A measurement method for the Brillouin scattering spectrum of seawater based on a Fizeau interferometer and multichannel PMT was established. According to the principle of equal inclination interference, equal-thickness RB interference fringes were produced using the Fizeau interferometer. After compression in the direction where light has the same frequency, the discrete scattering spectrum was quickly obtained using the multichannel PMT. With the spectrum reconstruction method, the Brillouin frequency shift and linewidth were obtained, and the temperature and salinity of the water could be retrieved.

The Brillouin frequency shift and linewidth error for different numbers of PMT channels were simulated and analyzed. The parameters of the Fizeau interferometer were determined based on the Brillouin spectrum and the interferometer characteristics. Considering the existing experimental instruments and cost, the number of PMT channels was set to 16. To verify the anti-jamming capability of the proposed method, the Brillouin shift and linewidth were retrieved by simulation when the spectrum was shifted. The results showed that the Brillouin frequency shift error was less than 2 MHz in the range of 0.762 GHz, and the Brillouin linewidth error was mostly less than 3 MHz. This demonstrated that the proposed method has strong anti-interference ability.

Moreover, a Brillouin energy experimental system was designed and built to analyze the detection performance of the proposed method for depth detection. The experimental results showed that the Brillouin scattering energy under experimental conditions accounted for 10–30% of the total scattering echo energy. The attenuation coefficient of the water was 0.099, and the maximum measurement depth was 13.68 m. Based on the analogy method, the maximum detection depth of the proposed system was found to be 42.90 m under the same attenuation coefficient.

In the end, a simulation analysis of the measurement depths under different SNRs was performed to further analyze the detection performance in terms of the accuracy. The results showed that under a condition where the temperature and salinity measurement accuracies were 0.5 ℃ and 1‰, respectively, the SNR of the scattered signal could not be lower than 35, and the effective detection distance for a single measurement was 27.73 m for a single measurement time. The depth display under different measurement times was calculated to analyze the influence of the cumulative number on the measurement depth. The effective detection depth for 100 laser pulses was 75.48 m, the effective detection depth for 1000 laser pulses was 97.71 m, and the ranger detection depth for 10,000 laser pulses was 119.28 m. The best cumulative pulse measurement time of 1000 was selected when integrating the effective measurement depth and time resolution. The measurement depths under different water quality parameters were analyzed. The results showed that the effective detection depth of the proposed method exceeded 40.51 m for various water bodies when an accumulation time of 10 s was adopted.

Compared with the current Brillouin spectral measurement methods, the proposed method offers a number of advantages. First, this method can obtain the Rayleigh Brillouin spectrum quickly compared with the spectral measurement method by using the FP interferometer. Second, as the response of the multichannel PMT is faster than that of ICCD, the proposed method can effectively overcome the disadvantage of the method by using FP etalon and ICCD in profile measurement. In addition, the completed Rayleigh–Brillouin scattering spectra can be acquired by the proposed method, the stability of the spectral measurement and the anti-interference ability are improved compared with the method based on the double-edge technology. Therefore, this method satisfies the requirements of real-time spectrum measurement, spectral integrity, continuity, and stability in ocean remote sensing.

Owing to the multibeam interference used in this system, a careful fabrication process for the Fizeau interferometer is required to ensure the integrity and accuracy of the Rayleigh–Brillouin spectra. Additionally, for the multichannel PMT, the response performance of each channel will not be completely consistent. Therefore, a nonuniform correction must be used for actual application. Hence, the actual performance of this method remains to be verified.

## 6. Conclusions

Considering the requirements of real-time spectrum measurement, spectral integrity, continuity, and stability, a measurement method based on a Fizeau interferometer and multichannel PMT is proposed to measure the temperature and salinity of seawater. In this paper, the design of a corresponding measurement system on an airborne platform is described. This method has strong anti-interference ability; under a temperature measurement accuracy of 0.5 °C and a salinity measurement accuracy of 1‰, the effective detection depth exceeds 40.51 m. The proposed method can meet the requirements for high-precision synchronous measurements of environmental element profiles of seawater. In future studies, a setup should be built for verification and analysis and our method will further promote the application of Brillouin lidar in the ocean.

## Figures and Tables

**Figure 1 sensors-23-00446-f001:**
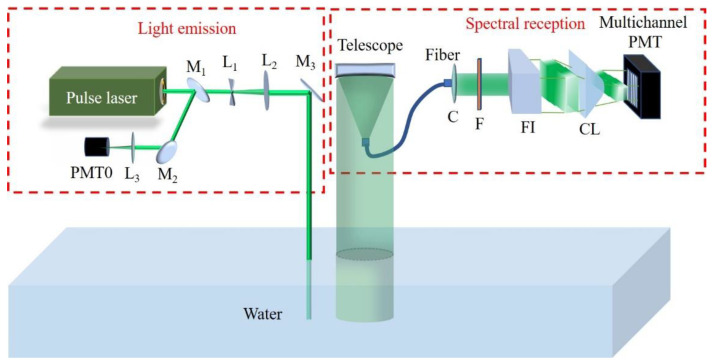
Schematic of the Brillouin spectral method based on a Fizeau interferometer and multichannel PMT. The light emitted by the pulse laser is split by M_1_ into two parts. One is collected by PMT0 as the reference beam to monitor the energy changes. The other part, after the beam expander system, directly enters the water to produce a scattering light. Subsequently, the 180° backscattered light is first collected by a Cassegrain telescope and transmitted through an optical fiber. After collimation, the light successively passes through a narrowband filter (F) and the Fizeau interferometer (FI), producing an interference fringe. The fringe is then sharpened by the cylindrical lens (CL) and finally detected by the multichannel PMT. (M_i_ and L*_i_* indicate the mirrors and lenses, respectively, and C is the fiber collimator).

**Figure 2 sensors-23-00446-f002:**
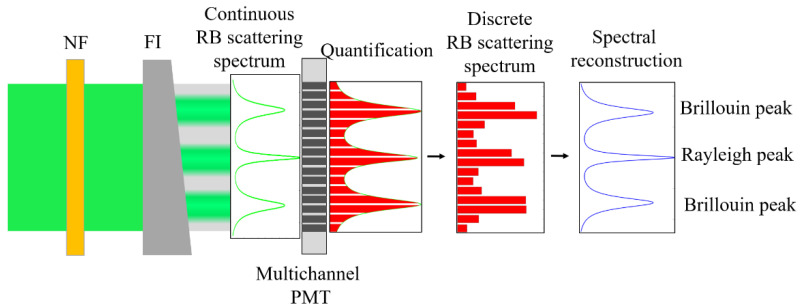
Schematic of data acquisition and spectral reconstruction.

**Figure 3 sensors-23-00446-f003:**
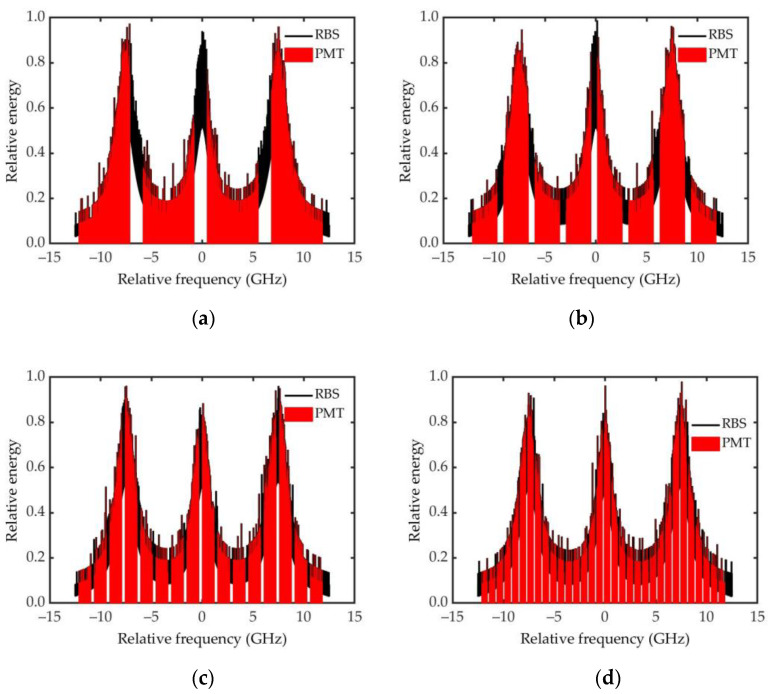
RB scattering spectrum obtained under different channel numbers of the PMT: (**a**) 4; (**b**) 8; (**c**) 16; and (**d**) 32.

**Figure 4 sensors-23-00446-f004:**
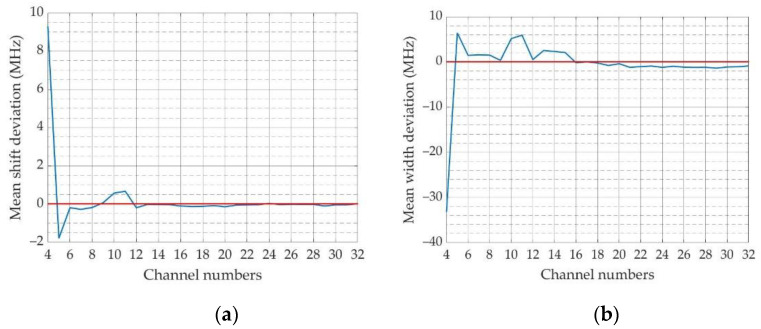
Deviations under different channel numbers of the PMT: (**a**) Brillouin shift; (**b**) Brillouin linewidth.

**Figure 5 sensors-23-00446-f005:**
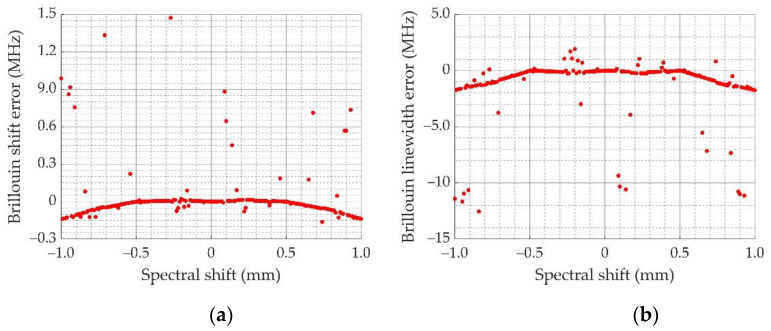
Result of spectral measurement accuracy considering spectral drift: (**a**) Brillouin frequency error retrieved when the spectra shift from −0.762 to 0.762 GHz; (**b**) Brillouin linewidth error retrieved when the spectra shift from −0.762 to 0.762 GHz.

**Figure 6 sensors-23-00446-f006:**
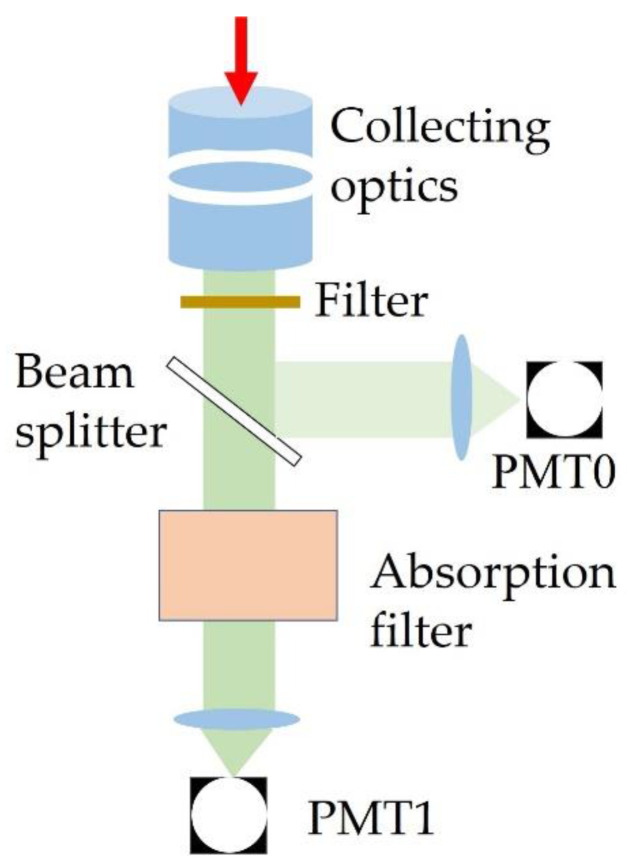
Schematic of the system used for Brillouin energy measurement.

**Figure 7 sensors-23-00446-f007:**
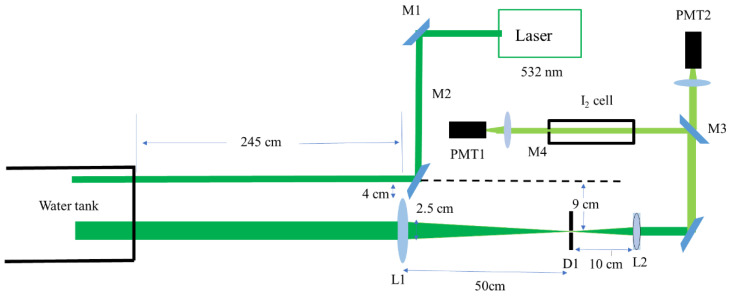
Diagram of the experimental system for Brillouin energy measurement in water.

**Figure 8 sensors-23-00446-f008:**
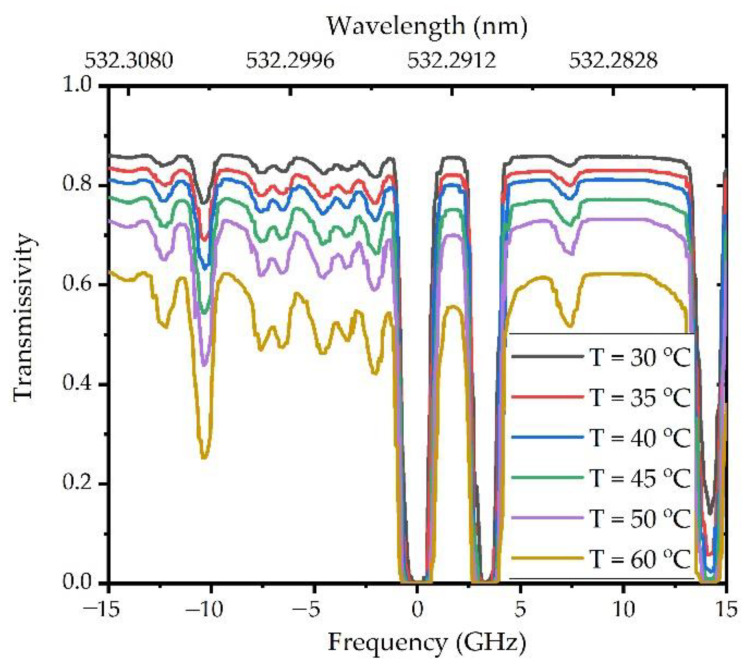
Iodine absorption lines under different temperatures within 15 GHz at a wavelength of approximately 532.293 nm.

**Figure 9 sensors-23-00446-f009:**
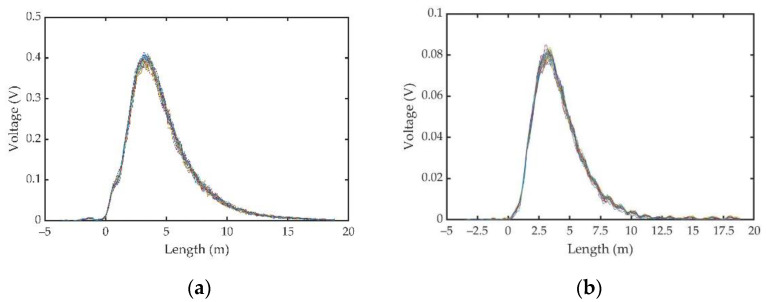
Scattered signals before and after passing through an iodine cell: (**a**) overall scattered signal before passing through the iodine cell; (**b**) Brillouin scattering signal after passing through the iodine cell.

**Figure 10 sensors-23-00446-f010:**
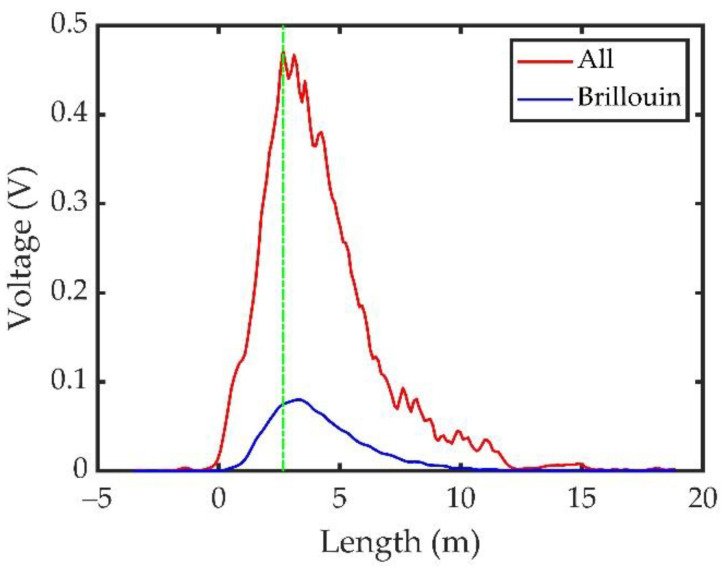
Results of the Brillouin energy measurement experiment before and after passing through the iodine cell after averaging 100 times.

**Figure 11 sensors-23-00446-f011:**
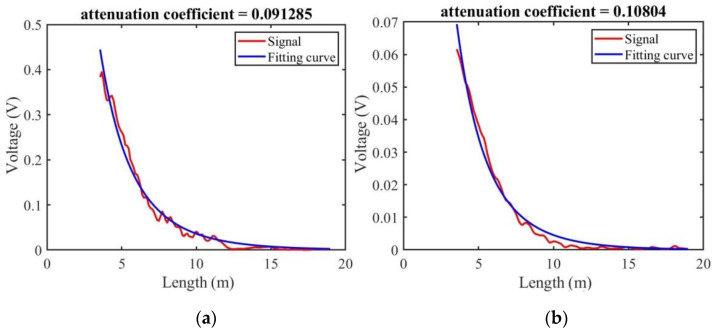
Calculation of the attenuation coefficient of water: (**a**) fitting result for the total scattering energy; (**b**) fitting result for the Brillouin scattering energy.

**Figure 12 sensors-23-00446-f012:**
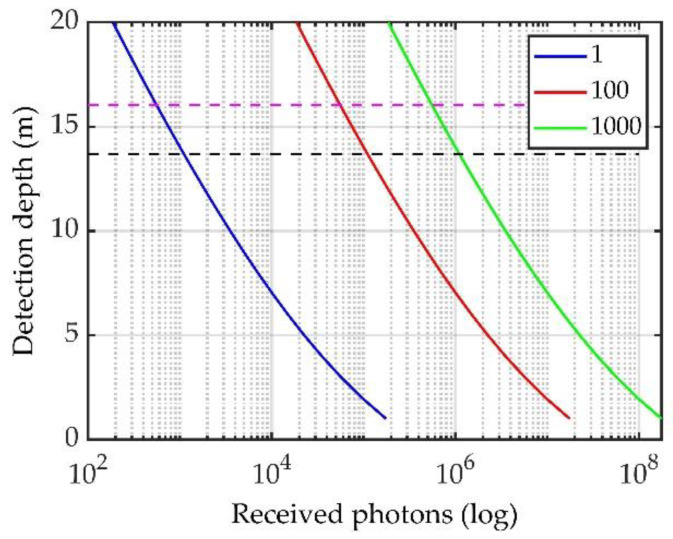
Variation in the number of Brillouin scattering photons with the measurement depth.

**Figure 13 sensors-23-00446-f013:**
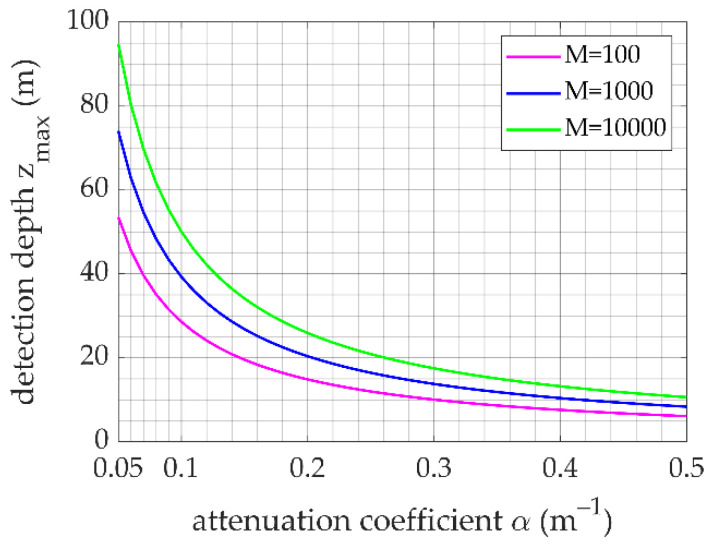
Maximum detection depth for different attenuation coefficients.

**Figure 14 sensors-23-00446-f014:**
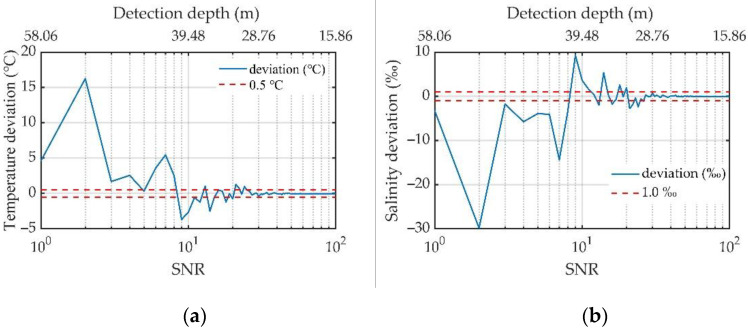
Relationships between the SNR and accuracy of the temperature and salinity measurements: (**a**) temperature deviation with respect to the SNR; (**b**) salinity deviation with respect to the SNR.

**Figure 15 sensors-23-00446-f015:**
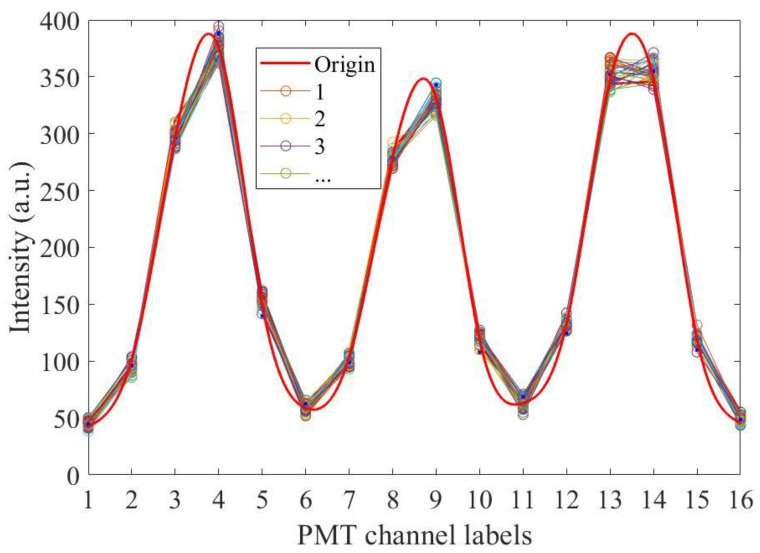
Simulated scattering spectrogram received by a 16-multichannel PMT under an SNR of 35.

**Figure 16 sensors-23-00446-f016:**
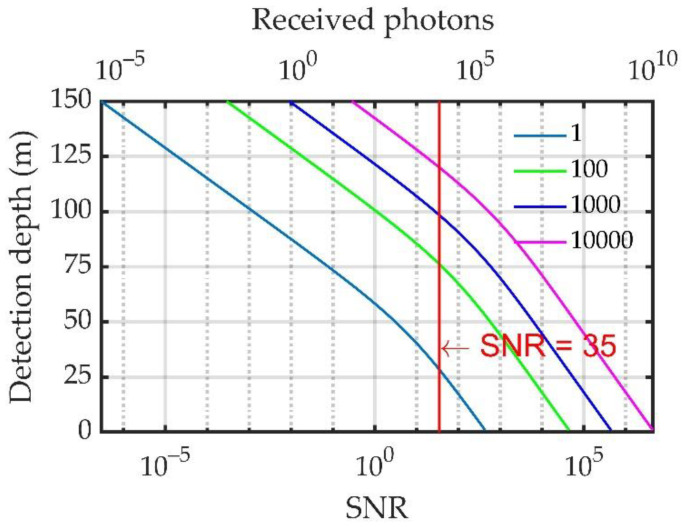
Relationship between the SNR and detection depth under different accumulation numbers.

**Figure 17 sensors-23-00446-f017:**
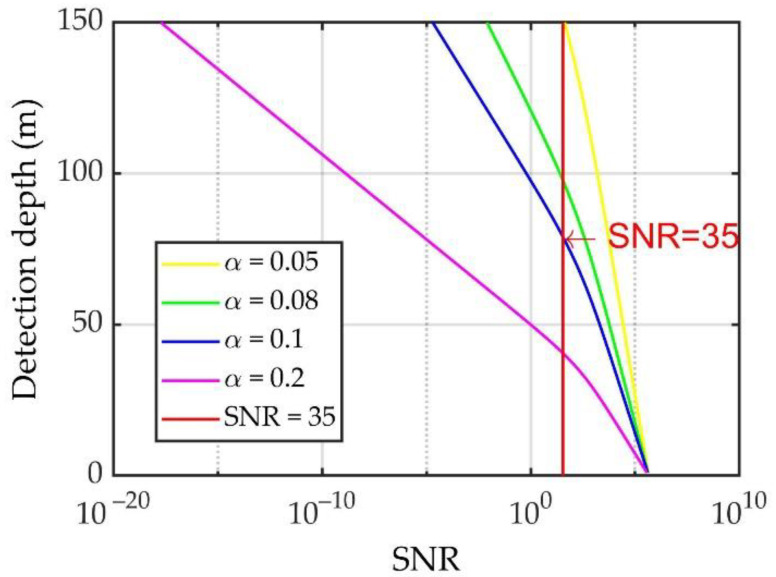
Effective detection depth under different water attenuation coefficients.

**Table 1 sensors-23-00446-t001:** Design parameters of the system.

Parameter	Value	Parameter	Value
Laser energy *E*	20 mJ	Pulse width τ	10 ns
Laser wavelength *λ*	532 nm	Brillouin backscattering coefficient $\sigma_{B}$	$2.4\times 10^{-4}$
Telescope radius *r*	0.05 m	Height *H*	100 m
Attenuation coefficient α	0.08	Atmospheric attenuation efficiency $\alpha_{A}$	0.81
Detection efficiency $\xi_{s}$	0.13	Optical efficiency $\xi_{l}$	0.4
Refractive index *n*	1.33	Repetitive frequency	100 Hz

**Table 2 sensors-23-00446-t002:** Parameters of the Brillouin energy measurement system.

Parameter	Value	Parameter	Value
Laser energy *E_0_*	0.2 mJ	Pulse width $\tau_{0}$	10 ns
Laser wavelength $\lambda_{0}$	532.293 nm	Repetitive frequency	100 Hz
Telescope diameter *d_0_*	0.025 m	Scattering angle *θ*	179°
Attenuation coefficient α	0.099	Height *H*_0_	2.45 m
Averaging number *M*_0_	100		

## Data Availability

The data presented in this study are available on request from the corresponding author.

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
