# Peer review of "Remote Sensing of Seawater Temperature and Salinity Profiles by the Brillouin Lidar Based on a Fizeau Interferometer and Multichannel Photomultiplier Tube"

_sensors, 2022, doi:10.3390/s23010446_

Round 1
Reviewer 1 Report
1. The abstract could be improved further. Typically, an abstract is a condensed version of the whole document and should be read independently (stand-alone). An abstract should begin with the significance of the research, then include the goals or purposes of the study, the procedures and methodologies used, the findings and conclusions, and finally any particular policy implications.
2. The introduction section is too big! Please precise it by focusing on the general background, rationale, research gaps, and objectives. Moreover, this paper's scientific contribution or novelty should be more clearly stated. What issue do the authors want to answer in the study, for example? What makes this study different from others? This information should be included in the introduction part.
3. The methodology section is exceptionally too big! The methodology section needs to be simplified and concise for better understanding by the readers. Author could add a conceptual framework for the claimed model. Moreover, a map of the study area with geographical data, and more details of the data used should be added.
4. The discussion chapter is lacking. There isn't any debate, and the conclusions are not backed up by literature. As a result, the authors should provide a discussion of the major findings as well as comparisons to prior research in the field. I would suggest writing a separate discussion section by separating Section 4 (Brillouin lidar system performance).
5. Since uncertainty is interlinked in modeling, what are the potential areas of uncertainty in the model used in this study, and how did these uncertainties of model results addressed?
6. The authors didn’t mention any limitations of the study, however, several limitations exist in the study. So, the authors are recommended to mention the potential limitation of the study in the discussion or conclusion section.
7. In conclusion, one paragraph should be based on the main conclusions and the second paragraph should state the policy suggestions along with the future research directions in a good order. Present policy suggestions are not very sound. Please provide a sound policy implications stemming from the main findings of this work.
8. Indications of future work may be pointed out to guide interested readers to the subsequent work.
9. The references should be checked for consistency of format.
Author Response
Thank you for your comments concerning our manuscript entitled ‘Remote sensing of seawater temperature and salinity profiles by the Brillouin lidar based on Fizeau interferometer and multi-channel photomultiplier tube’. The responds to the reviewer’s comments are as flowing:
Responds to the reviewer’s comments:
Reviewer: 1
Comments:
- The abstract could be improved further. Typically, an abstract is a condensed version of the whole document and should be read independently (stand-alone). An abstract should begin with the significance of the research, then include the goals or purposes of the study, the procedures and methodologies used, the findings and conclusions, and finally any particular policy implications.
Reply: We agree with the reviewer very much. We modified the structure of the writing. We highlighted the current shortcomings of the current Brillouin spectral measurement methods in real Brillouin lidar application. In addition, we have clearly explained the meaning of this article in the end that our method is a good choice for seawater remote sensing.
- The introduction section is too big! Please precise it by focusing on the general background, rationale, research gaps, and objectives. Moreover, this paper's scientific contribution or novelty should be more clearly stated. What issue do the authors want to answer in the study, for example? What makes this study different from others? This information should be included in the introduction part.
Reply: We sincerely thank the reviewer’s valuable feedback. We modified the introduction and deleted some redundant parts. In the last but two paragraphs, we show that our idea that a complete Rayleigh and Brillouin spectra can solve the problem of double edge method. Then in next paragraph we display our novelty method and show that how do we method the complete Rayleigh and Brillouin spectra. In addition, we state our purposed for this paper that to solve the shortage of the measurement method based on the double-edge technology in high-precision detection. Besides, we have stated the characteristic of each method of Brillouin lidar and the difference between the Fizeau method and other method is easy to distinguish.
- The methodology section is exceptionally too big! The methodology section needs to be simplified and concise for better understanding by the readers. Author could add a conceptual framework for the claimed model. Moreover, a map of the study area with geographical data, and more details of the data used should be added.
Reply: We feel great thanks for your professional review work on our article. This section contains two meaning actually: the system and the theory of the proposed method. To make this section more clearly, we added two subtitles. In the first and second paragraphs, we show the proposed system and want to tell readers how to get the complete Rayleigh Brillouin spectra with the system. The rest parts are data processing to display how to get the temperature and salinity based on the obtained complete Rayleigh Brillouin spectra. Moreover, we here just show a new method to measure the ocean temperature and salinity, and the system doesn't already exist. So, there is no need to add the geographical data. We think it can be added in future work.
- The discussion chapter is lacking. There isn't any debate, and the conclusions are not backed up by literature. As a result, the authors should provide a discussion of the major findings as well as comparisons to prior research in the field. I would suggest writing a separate discussion section by separating Section 4 (Brillouin lidar system performance).
Reply: Based on these comments and suggestions, we have made careful modifications to the original manuscript. We add the discussion part to discuss the detection accuracy of the Brillouin lidar. The discussion contains the several parts: the discussions of the performance of the proposed method, the advantages of the proposed method compared with the current Brillouin lidar technique and the limitations of the proposed method.
- Since uncertainty is interlinked in modeling, what are the potential areas of uncertainty in the model used in this study, and how did these uncertainties of model results addressed?
Reply: I don’t quite understand what the model means. In our paper, the retrieving model of temperature and salinity is used and the discussion of the model can be found in Refence [34]. Here, we don’t make a further analysis for the uncertainties in the model.
- The authors didn’t mention any limitations of the study, however, several limitations exist in the study. So, the authors are recommended to mention the potential limitation of the study in the discussion or conclusion section.
Reply: Thank you for your nice comments on our article. According to your suggestions, we added the limitations of the study in the discussion section.
- In conclusion, one paragraph should be based on the main conclusions and the second paragraph should state the policy suggestions along with the future research directions in a good order. Present policy suggestions are not very sound. Please provide a sound policy implications stemming from the main findings of this work.
Reply: We rewrite the conclusion and make it more concise. The conclusion is set as two paragraphs. First one is the main conclusion and the second is the future research directions.
- Indications of future work may be pointed out to guide interested readers to the subsequent work.
Reply: Thanks for your suggestions. We rewrite the part of the future work and shows people a clearer instruction
- The references should be checked for consistency of format.
Reply: We really appreciate your advice. We checked the references and made some modifications. Thank you very much for your help.
Reviewer 2 Report
This paper developed a new lidar system for spectrum measurement on an airborne platform
based on a Fizeau interferometer and multichannel photomultiplier tube. The lidar system uses
time-of-flight information to measure the depth and relies on Bril-louin scattering as the
temperature and salinity indicator.The system parameters were first optimized and analyzed.
Based on the analysis results, the performance of the system in terms of detection depth and
accuracy was evaluated. The results showed that this method has strong anti-interference
ability, and under a temperature measurement accuracy of 0.5 ℃ and a salinity measurement
accuracy of 1 ‰, the effective detection depth exceeds 40.51 m. These results indicate that the
proposed method performs well in seawater remote sensing.
The research method is credible and the technical route is reasonable.The proposed method can
meet the requirements of high-precision synchronous measurements of the environmental
element profiles of seawater.
Suggest:
1.P11 The temperature label should not affect the curve in Fig8
2.P15 The units marked on the vertical and horizontal coordinates should be consistent
with other graph representations.
Author Response
Thank you for your comments concerning our manuscript entitled ‘Remote sensing of seawater temperature and salinity profiles by the Brillouin lidar based on Fizeau interferometer and multi-channel photomultiplier tube’. The responds to the reviewer’s comments are as flowing:
Responds to the reviewer’s comments:
Comments:
- P11 The temperature label should not affect the curve in Fig8
Reply: We feel sorry for our carelessness. Based on these comments and suggestions, we changed the Figure 8 with moving the label to another location.
2.P15 The units marked on the vertical and horizontal coordinates should be consistent
with other graph representations.
Reply: We sincerely thank the reviewer for careful reading. We changed the format of the units marked on the Figure 13 and made it more consistent.
Round 2
Reviewer 1 Report
The authors improved their manuscript by making significant modifications in response to the reviewers' suggestions. Since I have no other concerns regarding the manuscript, I think it should be approved for publication in its current form.